

# The effects of acute hydrogen peroxide exposure on respiratory cilia motility and viability

Richard Francis

Biomedicine and Cell and Molecular Sciences; College of Public Health, Medical and Veterinary Science, James Cook University, Townsville, Queensland, Australia

## ABSTRACT

COVID-19 has seen the propagation of alternative remedies to treat respiratory disease, such as nebulization of hydrogen peroxide ($H_2O_2$). As $H_2O_2$ has known cytotoxicity, it was hypothesised that $H_2O_2$ inhalation would negatively impact respiratory cilia function. To test this hypothesis, mouse tracheal samples were incubated with different $H_2O_2$ concentrations (0.1–1%) then cilia motility, cilia generated flow, and cell death was assessed 0–120 min following $H_2O_2$ treatment. 0.1–0.2% $H_2O_2$ caused immediate depression of cilia motility and complete cessation of cilia generated flow. Higher $H_2O_2$ concentrations ($\geq$0.5%) caused immediate complete cessation of cilia motility and cilia generated flow. Cilia motility and flow was restored 30 min after 0.1% $H_2O_2$ treatment. Cilia motility and flow remained depressed 120 min after 0.2–0.5% $H_2O_2$ treatment. No recovery was seen 120 min after treatment with $\geq$1% $H_2O_2$. Live/dead staining revealed that $H_2O_2$ treatment caused preferential cell death of ciliated respiratory epithelia over non-ciliated epithelia, with 1% $H_2O_2$ causing 35.3 ± 7.0% of the ciliated epithelia cells to die 120 min following initial treatment. This study shows that $H_2O_2$ treatment significantly impacts respiratory cilia motility and cilia generated flow, characterised by a significant impairment in cilia motility even at low concentrations, the complete cessation of cilia motility at higher doses, and a significant cytotoxic effect on ciliated respiratory epithelial cells by promoting cell death. While this data needs further study using *in vivo* models, it suggests that extreme care should be taken when considering treating respiratory diseases with nebulised $H_2O_2$.

# INTRODUCTION

The 21st century has seen complementary and alternative medicine (CAM) surge in popularity, with CAM adoption in adults approaching 40% in the U.S. (*Ventola, 2010*), 60% in Australia (*von Conrady & Bonney, 2017*), and up to 40% in some European countries (*Kemppainen et al., 2018*). The increased acceptance of CAM is thought to be fuelled by the rise of alternative health/social media which can quickly overwhelm individuals with misinformation, conspiracy theories, and quackery (*Delgado-Lopez & Corrales-Garcia, 2018*; *Wu et al., 2022*). CAM associated misinformation has never been more evident than during the recent COVID-19 pandemic, where the usual rogues' gallery

Corresponding author
Richard Francis,
richard.francis@jcu.edu.au

of scientifically unproven treatments has been proposed to successfully treat the disease, including herbs, teas, essential oils, tinctures, vitamins, and products such as colloidal silver (*Jeon et al., 2022*). Nebulization and inhalation of hydrogen peroxide ($H_2O_2$) is another CAM treatment for COVID-19 which has been proposed by numerous sources, both within the CAM community itself (Table S1), but also by some within the mainstream medical community (*Caruso, Del Prete & Lazzarino, 2020a*, *Caruso et al., 2020b*; *Cervantes Trejo et al., 2021*).

$H_2O_2$ is a colourless liquid at room temperature with powerful oxidizing activity (*National Center for Biotechnology Information, 2022*) and is used in many industries as a general-purpose disinfectant or bleaching/whitening/deodorising product (*Watt, Proudfoot & Vale, 2004*). $H_2O_2$ also has well characterised antimicrobial and antiviral activity (*Heckert et al., 1997*; *McDonnell, 2004*), and has seen clinical use since the early 20th century as a popular topical antiseptic for wound irrigation or the sterilisation of instruments/surfaces (*Lu & Hansen, 2017*; *Watt, Proudfoot & Vale, 2004*). The utility of $H_2O_2$ as a surface disinfectant was recently shown when a 1-min application of 0.5% $H_2O_2$ efficiently disinfected inanimate surfaces of human coronaviruses (*Kampf et al., 2020*).

$H_2O_2$ is also a very caustic compound displaying a range of toxic effects on living cells and tissues (*Vilema-Enriquez et al., 2016*; *Watt, Proudfoot & Vale, 2004*). A small number of *in vitro* studies utilizing isolated airway tissue has revealed that $H_2O_2$ treatment negatively impacts normal respiratory epithelia function as characterised by a reduction in cilia beat frequency (CBF) and increased epithelial cell death (*Burman & Martin, 1986*; *Feldman et al., 1994*; *Honda et al., 2014*; *Jeppsson et al., 1991*; *Kakuta, Sasaki & Takishima, 1991*; *Kobayashi et al., 1992*; *Nakajima et al., 1999*). In addition, several clinical case reports have shown a connection between both acute and chronic $H_2O_2$ inhalation and the development of interstitial lung disease (*Kaelin, Kapanci & Tschopp, 1988*; *Manfra et al., 2021*; *Nguyen & Gordon, 2015*; *Riihimäki et al., 2002*). However, while the toxicity of $H_2O_2$ is well recognized within the scientific literature, it hasn't stopped the CAM community from stating $H_2O_2$ inhalation is "a completely non-toxic therapy" and suggesting "nebulisation can be administered as often as desired" to cure or prevent any number of respiratory related diseases (Table S1).

Thus, the aim of this study was to determine how acute $H_2O_2$ exposure effects the ciliated respiratory epithelia, then to assess the ability of the epithelia to recover from this initial $H_2O_2$ exposure. To assess this, $H_2O_2$ doses and treatment times were selected based on those currently recommended within the CAM literature (Table S1), which is much higher than examined in any previous study. Ciliated respiratory epithelia samples were then imaged at differing timepoints following initial $H_2O_2$ treatment and the number of epithelia cells with motile cilia, CBF, cilia generated flow, and epithelia cell survival were all quantified.

## MATERIALS AND METHODS

### Animals

All animal procedures were conducted in accordance with the James Cook University Animal Ethics Committee (Ethics# A2783). C57/BL6 mice of mixed sex and age destined

for euthanasia during routine colony maintenance were donated to this study by the Australian Institute of Tropical Health & Medicine small animal colony. Animals were housed in an air-conditioned room in racked mouse cages connected to a Smart Flow ventilation system (Tecniplast, Buguggiate, Italy). Animals had access to food (standard rodent chow) and water *ad libitum*. Enrichment of animal cages was achieved by including a range of bedding materials (sawdust and shredded paper), and cardboard tubes for mice to hide/sleep in. Mice were delivered weekly and were euthanised the same week using carbon dioxide asphyxiation. Euthanasia was confirmed by loss of corneal and toe pinch reflexes. All animals delivered for this study were euthanised and used for data collection.

## Ciliated epithelia isolation and preparation

Sections of whole-mount ciliated trachea epithelia were harvested, treated, and imaged in L-15 media lacking phenol red (21083027; ThermoFisher Scientific, Waltham, MA, USA) supplemented with 10% FBS (16000036; ThermoFisher Scientific, Waltham, MA, USA), 100 units/ml of penicillin G sodium, and 100 µg/ml of streptomycin sulfate (15140122; ThermoFisher Scientific, Waltham, MA, USA). Mice were euthanised *via* $CO_2$ asphyxiation and trachea were immediately isolated and placed into L-15 media on ice before being prepared for video microscopy as previously described (*Francis & Lo, 2013*). In brief, trachea segments were cut longitudinally through the middle of the trachealis muscle then mounted with ~100 µl of L-15 media in an imaging chamber constructed using two $24 \times 50$ mm #1 coverslips (EPBRCS24501GP; Bio-Strategy, Hobsonville, New Zealand) sandwiching a 0.254 mm thick silicone sheet gasket (CASS-.010X36-64909; AAA Acme Rubber Co, Tempe, AZ, USA). 1–2 drops of 0.50 µm microspheres (17152-10; Polysciences, Inc., Warrington, PA, USA) were added to ~4 ml of L-15 media used to mount trachea sections for tracking of cilia generated flow.

## $H_2O_2$ treatment

$H_2O_2$ doses and treatment times were based on recent CAM literature, with most recommending nebulization of 3% $H_2O_2$ for 10 to 15 min (*Cervantes Trejo et al., 2021*) (Table S1). L-15 media containing different $H_2O_2$ concentrations (0.1%, 0.2%, 0.5%, 1%) was freshly made before each experiment. Four trachea samples were collected from each animal and randomly assigned to either a sham treatment group, or one of the four $H_2O_2$ treatment concentrations. Sham treatment was L-15 media without added $H_2O_2$. Tissue samples were incubated for 10 min in their assigned treatment, then washed three times in fresh L-15 media before storage at 37 °C before imaging. Division of each trachea into four samples allowed tissue from the same animal to be imaged at four different timepoints (0, 30, 60, 120 min) following $H_2O_2$ or sham treatment.

## Imaging cilia motility

Samples were imaged using a Zeiss Axiovert 200 microscope with a 63x/1.4 Oil objective (420782-9900; Zeiss, Jena, Germany), Immersol 518F Immersion Oil (444960-0000; Zeiss, Jena, Germany), and DIC microscopy as previously described (*Scopulovic et al., 2022*). In brief, samples were imaged at 37 °C and recordings of cilia motility were collected using

a Sony Exmore CMOS sensor (EM101500A; ProSciTech, Kirwan, QLD, Australia). One second movies (AVI; uncompressed) were collected at ~300 fps for quantification of CBF, while 20 s movies (mp4; HEVC) were collected at 30 fps for quantification of motile cilia percent and cilia generated flow.

## Quantifying cilia motility and cilia generated flow

The impact of $H_2O_2$ treatment on cilia motility was first assessed by calculating the percent of motile ciliated cells (%MC) visible in each 20 s movie field of view, this was done by dividing the number of ciliated cells with motile cilia by the total number of ciliated cells visible (both motile and non-motile).

To measure cilia beat frequency (CBF), kymographs of motile cilia were generated from each 1 s (300 fps) movie in ImageJ (FIJI 2.3.0/1.53f) (*Schindelin et al., 2012*) as previously described (*Scopulovic et al., 2022*). CBF was then quantified from the kymographs using a custom MATLAB script (version 9.9.0, R2020b; MathWorks Inc, Natick, Massachusetts, USA).

Cilia generated flow was quantified from each 20 s movie (30 fps) using the Manual Tracking plugin for ImageJ (https://imagej.nih.gov/ij/plugins/track/track.html) to track the velocity of the 0.50 μm microspheres added to the bathing media (Movie S2). Cilia generated flow was assessed by tracking microsphere movement within the bathing media *via* two parameters, microsphere velocity and microsphere directionality. Microsphere directionality was calculated from the velocity data using Microsoft excel by dividing net microsphere displacement over time by total distance travelled (*i.e.*, was microsphere motion random or more directed). Microspheres moving in a straight-line display directionality ≈1; microspheres moving randomly display directionality ≈0. Thus, microsphere velocity reflected microsphere speed, while microsphere directionality reflected the linearity of microsphere movement.

Microspheres moving within empty imaging chambers were also tracked to determine Brownian motion values (Fig. S1; Movie S3). Values for Brownian motion were essential for proper interpretation of microsphere movement caused by cilia motility (or lack thereof).

## Quantification of epithelial damage by fluorescent microscopy

$H_2O_2$ induced cytotoxicity on respiratory epithelia samples was assessed using live/dead staining, immunohistochemistry, and fluorescent microscopy. Tracheal samples were stained using a Live/Dead Fixable Violet Dead Cell Stain Kit (L34963; ThermoFisher, Waltham, MA, USA) as per manufacturer instructions 120 min after treatment (Sham or $H_2O_2$). In brief, tracheal samples where washed three times in PBS, then incubated for 30 min at room temperature in a 1:1,000 dilution of the blue-fluorescent reactive dye in PBS. Samples were then washed once in PBS before being fixed in 4% PFA for 15 min. After fixation samples were washed three times in PBS, permeabilized for 10 min in PBST (0.2% Triton X-100 in PBS), blocked for 1 h in PBSGS (PBS + 5% Goat Serum), then incubated for 2 h at room temperature with a mouse anti-acetylated tubulin antibody (T7451; Sigma, Kawasaki, Japan) diluted 1:500 in antibody dilution buffer (PBS + 5% Goat Serum, + 0.1%

Triton X-100). Secondary fluorescent labelling was subsequently performed by incubating samples for 1 h with a goat anti-mouse FITC-conjugated antibody (115-095-003; Jackson ImmunoResearch Laboratories, West Grove, PA, USA) diluted in PBST containing a 1:1,000 dilation of TRITC-conjugated phalloidin (P1951; Sigma, Kawasaki, Japan). Trachea sections were mounted lumen side up on glass slides (7105-PPA; Livingstone, London, UK) with a drop of mounting media (F6182, Fluoroshield; Sigma, Kawasaki, Japan) under a 24 × 32 mm #1 coverslip (EPBRCS24321GP; Bio-Strategy PTY Ltd., Hobsonville, New Zealand) and 0.127 mm thick gasket cut from silicone sheet (CASS-.005X24-65908; AAA Acme Rubber Co, Tempe, AZ, USA) to prevent whole-mount samples from being crushed. Coverslip edges were sealed using clear nail polish (quick dry top coat; Revlon, Manhattan, NY, USA) and stored at 4 °C until imaged.

Fluorescently stained samples were imaged on a Zeiss LSM 710 confocal microscope using a 40x oil objective (Zeiss EC Plan-Neofluar 40x/1.30 Oil DIC M27) and ZEN black software (2.31 SP1). The Live/Dead fluorescent stain was imaged using 405 nm excitation (Laser Diode 405–30) and 494–552 nm emission spectra. FITC fluorescence was imaged using 488 nm excitation (Argon laser) and 494–552 nm emission spectra. TRITC fluorescence was imaged using 561 nm excitation (DPSS 561-10 laser) and 566–669 nm emission spectra. Samples were imaged using the line sequential scanning mode and z-stacking. Blue (Live/Dead violet) and orange (TRITC) fluorescence was imaged simultaneously followed by green (FITC) fluorescence. Image z-stack resolution was 1024/1024/~30 pixels (x/y/z), equating to pixel sizes of 0.21 μm/pixel (x/y), and 0.4–0.5 μm/pixel (z).

Respiratory epithelia cell damage was quantified from the collected fluorescent images using ImageJ (FIJI 2.3.0/1.53f) (*Schindelin et al., 2012*). The three colour channels in each z-stack image were collapsed using max intensity projection then combined to generate a single three colour image. The ImageJ 'Multi-point' tool was then used to count the total number of epithelial cells (TRITC-conjugated phalloidin), the total number of ciliated epithelial cells (FITC-labelled acetylated tubulin), the total number of dead epithelial cells (Live/Dead violet-stained cells without FITC-labelled cilia), and the total number dead ciliated epithelial cells (Live/Dead violet-stained cells with FITC-labelled cilia).

### Data analysis and statistics

Multiple measurements (≥3) of each parameter (%MC, CBF, flow velocity, flow directionality, live/dead counts) were made in each tissue sample for each treatment and each timepoint; resultant averages were then compared using two-way ANOVA and *post-hoc* Šídák's multiple comparisons test (Prism 9; GraphPad Software, San Diego, CA, USA). $P > 0.05$ was considered significant.

## RESULTS

A total of 40 mice were used for this study. Trachea from 30 mice were used for microscopy assessment of cilia motility and cilia generated flow, where each trachea was cut into four sections and assigned to a different treatment/time group providing an $n = 6$ for each treatment (Sham or $H_2O_2$) at each time point (0, 30, 60, 120 min).

For quantification of epithelial damage by fluorescent microscopy, trachea from 10 mice were subdivided to provide an $n = 6$ for each treatment (Sham or $H_2O_2$) at the 120-min time point.

## Cilia motility: control values following sham treatment

Sham treated tissue displayed constant well maintained cilia motility and cilia generated flow at all time points imaged (Figs. 1 and 2; Movie S1). The proportion of ciliated epithelial cells with motile cilia remained ≥97% in control samples (Figs. 1A and 2A), while CBF remained constant at ~20 Hz at all time points imaged (0–120 min) (Figs. 1B and 2B). Cilia generated flow remained constant at ~33 μm/sec (Figs. 1E and 2C), while flow directionality remained high (~0.9) (Figs. 1F and 2D) in all control samples at all time points indicating linear fluid flow across the ciliated epithelia was well maintained following sham treatment.

## Acute cilia motility response to $H_2O_2$ treatment

The immediate effect of $H_2O_2$ treatment on respiratory cilia was a significant impairment in cilia motility and cilia generated flow (Fig. 1; Movie S4). All $H_2O_2$ treatment concentrations produced a significant decrease in %MC (Fig. 1A) from the control value of $98.0 \pm 3.7\%$ to $34.1 \pm 34.1\%$ ($P < 0.0001$) in 0.1% $H_2O_2$ treated tissue and a further significant decrease to $5.4 \pm 7.3\%$ ($P < 0.0001$) in 0.2% $H_2O_2$ treated tissue. Higher $H_2O_2$ treatments (0.5% & 1%) both caused an immediate cessation ($P < 0.0001$) in all cilia motility (Fig. 1A; Movie S4). All $H_2O_2$ treatments also resulted in an immediate significant impairment in CBF, from $22.3 \pm 3.9$ Hz in control samples to $4.9 \pm 1.2$ Hz ($P < 0.0001$) in 0.1% $H_2O_2$ treated tissue and $4.9 \pm 0.4$ Hz ($P < 0.0001$) in 0.2% $H_2O_2$ treated tissue (Fig. 1B). CBF values were only calculated for cilia that displayed movement, except for the highest $H_2O_2$ concentrations (0.5% & 1%) which caused a complete cessation in all cilia motility, for which CBF was set as $0.0 \pm 0$ Hz (Fig. 1B).

Cilia generated flow was assessed by quantifying microsphere movement across the surface of the ciliated epithelium. Figure 1 shows representative microsphere traces following sham and 1% $H_2O_2$ treatment respectively (Figs. 1C and 1D; Movie S2). All $H_2O_2$ treatments resulted in a significant decrease ($P < 0.0001$) in cilia generated flow velocity from a control value of $32.6 \pm 8.4$ μm/sec to $7.3 \pm 4.1$ μm/sec in 0.1% $H_2O_2$ treated tissue, $5.7 \pm 1.1$ μm/sec in 0.2% $H_2O_2$ treated tissue, $6.5 \pm 0.9$ μm/sec in 0.5% $H_2O_2$ treated tissue, and $5.8 \pm 1.8$ μm/sec in 1% $H_2O_2$ treated tissue (Fig. 1E). Flow velocity in all $H_2O_2$ treated tissues was not significantly different from each other ($P > 0.05$) and were also not significantly different ($P > 0.05$) from the values obtained for microspheres moving in the absence of ciliated tissue *via* Brownian motion (Red dotted line in Fig. 1E). Flow directionality was also significantly reduced ($P < 0.0001$) in all $H_2O_2$ treatment groups (Fig. 1F) from a control value of $0.89 \pm 0.04$ to $0.40 \pm 0.21$ in 0.1% $H_2O_2$ treated tissue, $0.20 \pm 0.06$ in 0.2% $H_2O_2$ treated tissue, $0.21 \pm 0.07$ in 0.5% $H_2O_2$ treated tissue, and $0.22 \pm 0.05$ in 1% $H_2O_2$ treated tissue. Flow directionality in all $H_2O_2$ treated tissues was not

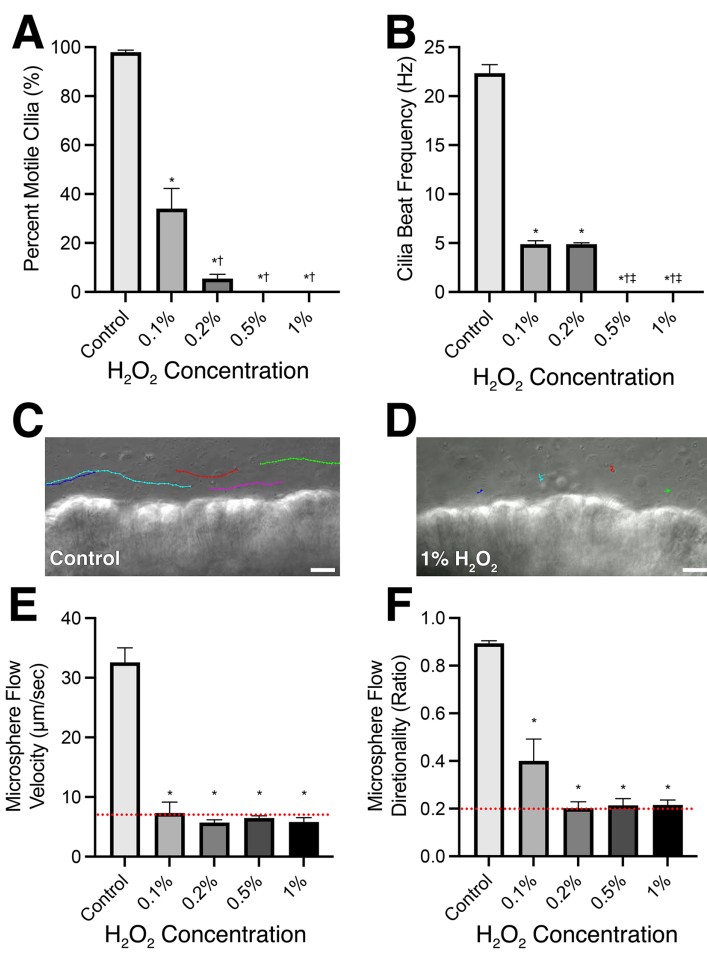

**Figure 1 Mouse respiratory cilia motility and cilia generated flow immediately following 10-min incubation in varying concentrations of $H_2O_2$.** (A) Percent of motile cilia (%MC) observed in tissues visualized. (B) Cilia beat frequency of the motile cilia visualized. (C) Representative microsphere tracks (different colour for each tracked microsphere) within L-15 media bathing control samples highlighting presence of cilia generated flow. (D) Representative microsphere tracks (different colour for each tracked microsphere) within L-15 media bathing 1% $H_2O_2$ treated samples highlighting absence of cilia generated flow. Quantification of microsphere velocity (E) and directionality (F) within bathing media to assess cilia generated flow. NB: Directionality was calculated by dividing net microsphere displacement by total distance travelled; microspheres moving in a straight-line display directionality ≈1; microspheres moving randomly display directionality ≈0. Red dotted lines represent the values calculated for microspheres moving *via* Brownian motion in an empty dish (*i.e.*, complete lack of cilia generated flow). Scale bars = 10 μm. Data presented as Mean ± SEM ($n ≥ 6$ for each data point). *Significantly different from control value ($P < 0.0001$), †Significantly different from 0.1% $H_2O_2$ dosed group ($P < 0.001$), ‡Significantly different from 0.2% $H_2O_2$ dosed group ($P < 0.0001$).

significantly different from each other ($P > 0.05$) or values obtained for microspheres moving in the absence of ciliated tissue *via* Brownian motion (Red dotted line in Fig. 1F). However, flow directionality following 0.1% $H_2O_2$ treatment showed a trend for higher directionality suggesting linear flow (albeit very slow) was maintained in a subset of these samples.

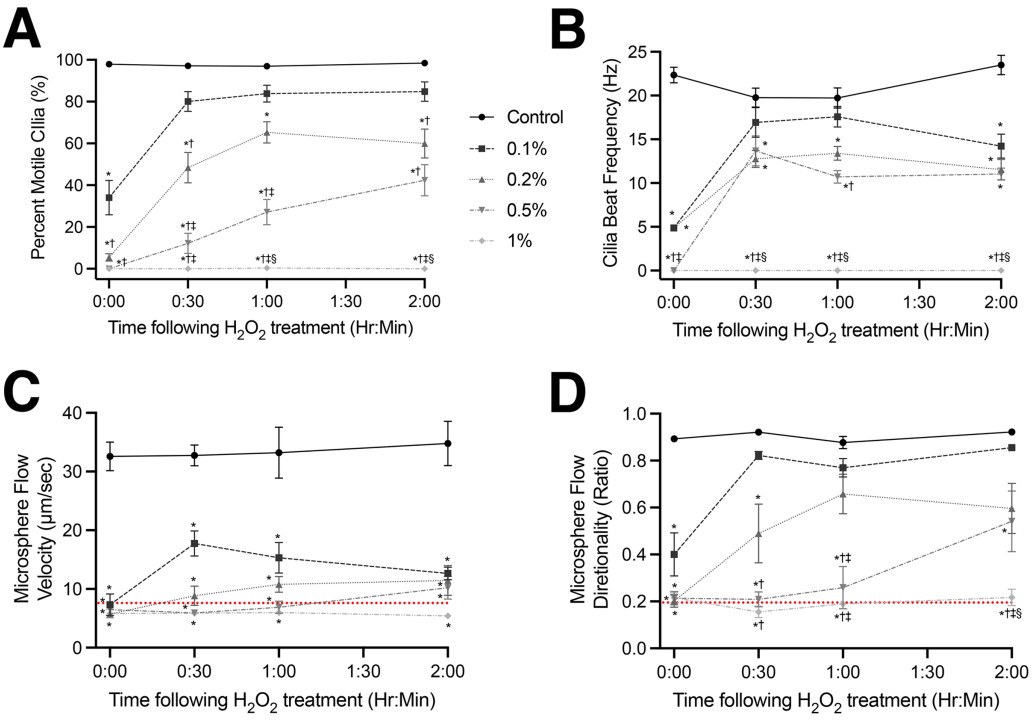

**Figure 2 Recovery of cilia motility and cilia generated flow following initial $H_2O_2$ treatment.**
(A) Percent of motile cilia (%MC) observed in tissues visualized. (B) CBF of motile cilia in tissues visualized. Tracking of microsphere velocity (C) and directionality (D) within bathing media to assess recovery of cilia generated flow. Red dotted lines represent the values calculated for microspheres moving *via* Brownian motion in an empty dish (*i.e.*, complete lack of cilia generated flow). Data presented as Mean ± SEM ($n \geq 6$ for each data point). *Significantly different from time matched control value ($P < 0.0001$), †Significantly different from time matched 0.1% $H_2O_2$ dosed group ($P < 0.01$), ‡Significantly different from time matched 0.2% $H_2O_2$ dosed group ($P < 0.01$), §Significantly different from time matched 0.5% $H_2O_2$ dosed group ($P < 0.01$).

## Recovery of cilia motility following $H_2O_2$ treatment

Recovery of cilia motility and cilia generated flow following $H_2O_2$ treatment was found to be treatment specific (Fig. 2; Movies S5–S7). While trending lower than sham treated animals, 30 min after the lowest $H_2O_2$ dose (0.1%) %MC had recovered to 80.1 ± 16.5% *vs* 97.2 ± 4.6% in controls and was not significantly different to the control values ($P > 0.05$) at the later timepoints (Fig. 2A). Conversely, while there was a graded recovery in %MC following 0.2% and 0.5% $H_2O_2$ treatment, %MC remained significantly lower compared to both control ($P < 0.0001$) and 0.1% $H_2O_2$ ($P < 0.01$) treatment groups (Fig. 2A). There was no recovery of %MC observed in airway tissues treated with the highest $H_2O_2$ dose (1%) at any timepoint (Fig. 2A). A total of 1% $H_2O_2$ treatment caused the complete cessation of all cilia motion which did not return after 120 min (Fig. 2A; Movie S7).

CBF displayed a similar recovery profile following $H_2O_2$ treatment. Namely, while trending lower than sham treated animals, 30 min after the lowest $H_2O_2$ dose (0.1%) CBF had recovered to 16.9 ± 5.9 Hz *vs* 19.7 ± 4.9 Hz in controls and was not significantly different from control values ($P > 0.05$) at 30- and 60-min post-treatment (Fig. 2B). However, after 120 min the 0.1% $H_2O_2$ treatment group displayed a significant drop in

CBF to $14.2 \pm 4.8$ Hz *vs* $23.5 \pm 4.5$ Hz in the control group ($P < 0.0001$). There was also a partial recovery in CBF following 0.2% and 0.5% $H_2O_2$ treatment after 30 min (Fig. 2B). CBF remained significantly lower following 0.2% and 0.5% $H_2O_2$ treatment *vs* control values ($P < 0.0001$) at all timepoints, but while these CBF values trended lower they were not significantly different from each other or the 0.1% $H_2O_2$ ($P < 0.01$) treatment group (Fig. 2B). No recovery in CBF was observed in airway tissues treated with the highest $H_2O_2$ dose (1%) at any timepoint (Fig. 2B).

Whereas %MC and CBF displayed graded recoveries depending on $H_2O_2$ treatment dose, cilia generated flow velocity, as assessed by microsphere tracking, showed a significant impairment in all $H_2O_2$ treatment groups at all timepoints (Fig. 2C). While samples 30 min after the lowest $H_2O_2$ dose (0.1%) did display a small improvement in cilia generated flow velocity to $17.8 \pm 4.7$ μm/sec *vs* $32.8 \pm 4.7$ μm/sec in controls, this change was not significantly different from the other treatment groups ($P > 0.05$), or the values obtained for microspheres moving in the absence of ciliated tissue *via* Brownian motion (red dotted line in Fig. 2C).

While cilia generated flow velocity remained depressed, linear flow as assessed by microsphere directionality was seen to return in a graded manner across the surface of the ciliated epithelium depending on $H_2O_2$ treatment concentration (Fig. 2D). Thirty minutes after the lowest $H_2O_2$ treatment (0.1%) linear flow had recovered to $0.82 \pm 0.03$ which was not significantly different from the control value of $0.92 \pm 0.02$ ($P > 0.05$) at all subsequent timepoints (Fig. 2D). Linear flow (Fig. 2D) also recovered in the 0.2% and 0.5% $H_2O_2$ treatment groups but remained significantly lower compared to control values ($P < 0.0001$) following 0.2% $H_2O_2$ treatment, and significantly lower compared to both control ($P < 0.0001$) and 0.1% $H_2O_2$ treatment ($P < 0.01$) following 0.5% $H_2O_2$ treatment (Fig. 2D). There was no recovery of linear flow observed in airway tissues treated with the highest $H_2O_2$ dose (1%) at any timepoint, which was not significantly different from the values obtained for microspheres moving in the absence of ciliated tissue *via* Brownian motion ($P > 0.05$) (red dotted line in Fig. 2D).

A small number of tissues were examined 0–120 min following 2% ($n = 2$) and 3% ($n = 2$) $H_2O_2$ treatment and were found to give identical results as seen following 1% $H_2O_2$ treatment, *i.e.*, a complete cessation of all cilia motion and cilia generated flow which did not return after 120 min (data not shown).

## Viability of tracheal epithelia cells following $H_2O_2$ treatment

Live/dead staining was used to assess the viability of tracheal epithelia cells 120 min following sham or $H_2O_2$ treatments (Fig. 3). Tracheal epithelia of control samples displayed a regular cobblestone arrangement which became more disorganized in samples treated with higher $H_2O_2$ concentrations (Fig. 3A). Cell counting revealed that all samples displayed the same proportion of ciliated epithelia cells *vs* non ciliated epithelia cells, with ~60% of mouse tracheal epithelia being non-ciliated *vs* ~40% ciliated (Fig. 3B). A dose-response trend for elevated cell death following treatment with higher $H_2O_2$ concentrations was observed (Fig. 3C). While both non-ciliated and ciliated tracheal epithelia cells showed a graded increase in cell death following increased $H_2O_2$

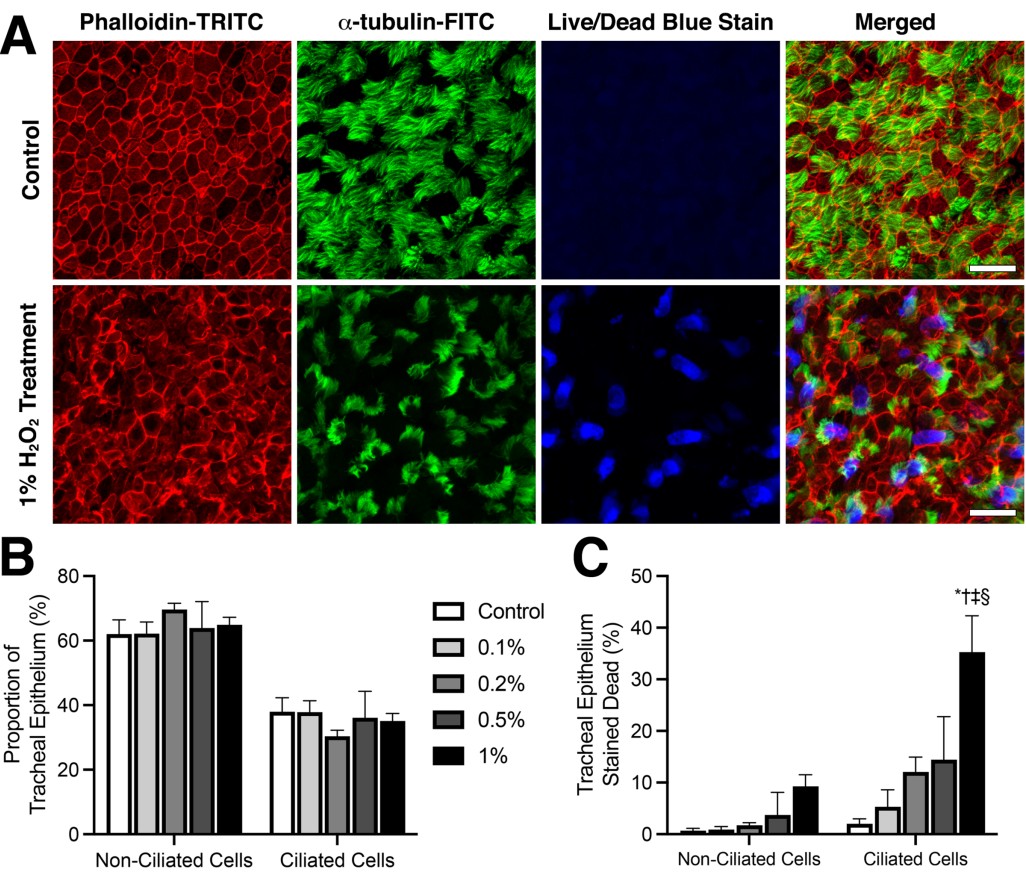

**Figure 3 Viability of ciliated tracheal epithelia cells following H₂O₂ treatment.** (A) Representative example of cell death observed within trachea epithelial following 10 min treatment with 1% $H_2O_2$ compared to control. Red: Phalloidin-TRITC fluorescence highlighting cell membranes; green: a-tubulin-FITC fluorescence highlighting ciliated cells; blue: fixable blue fluorescence live/dead stain highlighting dead epithelial cells. (B) Proportion of ciliated *vs* non-ciliated trachea epithelial cells in fluorescently stained tissues. (C) Quantification of cell death within ciliated *vs* non-ciliated trachea epithelial cells following 10-min $H_2O_2$ treatment. Data presented as Mean ± SEM. *Significantly different from control value ($P < 0.0001$), †Significantly different from 0.1% $H_2O_2$ dosed group ($P < 0.001$), ‡Significantly different from 0.2% $H_2O_2$ dosed group ($P < 0.0001$), §Significantly different from time matched 0.5% $H_2O_2$ dosed group ($P < 0.01$).

concentration, ciliated epithelia cells appeared more sensitive to $H_2O_2$ induced toxicity than non-ciliated cells (Fig. 3C). As highlighted by 1% $H_2O_2$ treatment which resulted in 35.3 ± 7.0% of the ciliated epithelia cells staining dead *vs* 2.0 ± 0.9% in control samples ($P < 0.0001$), whereas only 9.3 ± 2.3% of non-ciliated epithelia cells stained dead following 1% $H_2O_2$ treatment *vs* 0.7 ± 0.4% in control samples ($P > 0.05$) (Fig. 3C).

## DISCUSSION

The present study provides the first comprehensive overview of the toxic effect of $H_2O_2$ on the ciliated respiratory epithelium using $H_2O_2$ doses and treatment times recommended by CAM practitioners. To accomplish this, dose response-response assays were conducted on isolated mouse trachea tissue subjected to a single $H_2O_2$ treatment, then cilia beat

frequency, cilia generated flow, and cell death were all assessed, both directly after $H_2O_2$ treatment or following up to 2 h recovery.

## Control cilia motility data

The control trachea tissue samples displayed identical cilia activity across all the time points examined (0–120 min), as characterised by ~100% of cilia imaged displaying motility, CBF being maintained at ~20 Hz, cilia generated flow being maintained at ~35 µm/sec, and minimal to no respiratory epithelial cell death after 2 h in culture. This control cilia activity is consistent with previously published data from our laboratory (*Scopulovic et al., 2022*). It should be noted that a large heterogeneity exists within the published literature when reporting respiratory cilia motility in control samples. CBF is the most commonly assessed parameter (often the only assessed parameter) in respiratory cilia studies, probably due to the relatively simple nature of its measurement. Respiratory cilia studies report a broad range of control CBFs, ranging between 4–25 Hz (*Jing et al., 2017*; *Yasuda et al., 2020*; *Zahid et al., 2020*). While animal model may influence reported CBF, it's more likely this heterogeneity is caused by the different experimental protocols used, including different culture media, imaging modalities, and sample/temperature setups. However, while CBF is the most quantified parameter, CBF may be fairly meaningless by itself. The role of cilia in the lungs is to generate mucus flow across the surface of the ciliated epithelium to remove mucus and trapped contaminants, to accomplish this cilia need to beat in a coordinated manner using an optimal beat pattern (*Bustamante-Marin & Ostrowski, 2017*; *Raidt et al., 2014*), and previous studies have shown situations where CBF is maintained (or even elevated), but cilia generated flow is impaired due to defects in cilia beat pattern (*Chilvers, Rutman & O'Callaghan, 2003*; *Raidt et al., 2014*). Thus, the added impact offered by this study which not only assesses CBF, but also cilia generated flow.

## Immediate effect of single $H_2O_2$ exposures on respiratory cilia motility

The current study clearly demonstrates that $H_2O_2$ has an immediate negative effect on respiratory cilia function, with ≥1% $H_2O_2$ causing an immediate cessation of all cilia motion and the complete inhibition of cilia generated flow. Lower $H_2O_2$ concentrations also dramatically impacted cilia activity, resulting in significant reductions in CBF even at the lowest $H_2O_2$ concentration assessed (0.1%). Furthermore, while some cilia activity was maintained at the lowest $H_2O_2$ dose, cilia generated flow as assessed by tracking microsphere movement was significantly impaired, as highlighted by a reduction in linear flow and flow velocity which was slowed to the same levels as observed during simple Brownian motion (Fig. S1).

Only a handful of previous studies have examined the effect of $H_2O_2$ on respiratory cilia activity (*Burman & Martin, 1986*; *Feldman et al., 1994*; *Honda et al., 2014*; *Kakuta, Sasaki & Takishima, 1991*; *Kobayashi et al., 1992*; *Nakajima et al., 1999*). One major difference between this study and past studies is that they all utilized $H_2O_2$ concentrations at least 10x lower than the lowest dose used in the current study, and ~300x lower than the 3% concentration recommended by CAM practitioners (Table S1). Another major difference between this study and past studies is the wide variety of animal tissues used to model the

respiratory cilia, which includes airway tissue isolated from rats (*Burman & Martin, 1986*), guinea pigs (*Kakuta, Sasaki & Takishima, 1991*), sheep (*Kobayashi et al., 1992*), humans (*Feldman et al., 1994*; *Honda et al., 2014*), and bovines (*Nakajima et al., 1999*). $H_2O_2$ treatment times also varied greatly between studies, ranging from 10–15 min (*Burman & Martin, 1986*; *Kakuta, Sasaki & Takishima, 1991*) up to 24 h of continuous $H_2O_2$ treatment (*Honda et al., 2014*; *Nakajima et al., 1999*). However, while direct comparisons between this study and past studies may be difficult due to these major differences in experimental designs, all studies agree on one important point, namely, that respiratory epithelia exposure to $H_2O_2$ causes significant impairment to cilia motility and CBF. The power of this study is that it directly tests $H_2O_2$ concentrations and treatment times matching CAM recommendations, thus provides a more accurate appraisal of what may be occurring to the respiratory cilia of people following CAM advice.

Only one previous study assessed the effect of $H_2O_2$ on respiratory cilia motile function beyond simple CBF measurements. *Honda et al. (2014)* found that 24-h treatment of human bronchial cultures with 500 μM $H_2O_2$ caused a significant reduction in cilia generated flow as assessed by tracking migration rates of fluorescent microspheres. While the experimental design of the *Honda et al. (2014)* study was quite different from the current study, it does support this study finding $H_2O_2$ having a negative impact on cilia generated flow. Peculiarly, *Honda et al. (2014)* didn't measure CBF, so comparisons between CBF and cilia generated flow in their study is not possible.

## Recovery of cilia motility following $H_2O_2$ exposure

The current study monitored recovery of cilia motility up to 2 h following $H_2O_2$ treatment and found that respiratory cilia regained nearly normal motile function 30 min after treatment with the lowest $H_2O_2$ dose (0.1%), as assessed by recovery of %MC and CBF. Conversely, while some recovery in cilia motile function was observed after 0.2–0.5% $H_2O_2$ treatment (in a dose-response manner), cilia motility never recovered to pre-exposure levels, and remined impaired until the end of observation (2 h). This $H_2O_2$ toxicity was best highlighted following treatment with $H_2O_2$ concentrations ≥1% which resulted in no recovery of cilia motility at any time point assessed, with cilia remaining completely immotile until the end of the observation period. Only one previous study has attempted to assess the recovery of respiratory cilia motility following an initial $H_2O_2$ exposure. *Kakuta, Sasaki & Takishima (1991)* treated isolated guinea pig trachea rings for 15 min with 2 mM $H_2O_2$ and found $H_2O_2$ caused an initial ~60% reduction in CBF compared to controls, which then recovered to control levels 15 min after the $H_2O_2$ was removed. However, this recovery was dependant on the presence of a surfactant, and full recovery of CBF was not seen in samples without surfactant (*Kakuta, Sasaki & Takishima, 1991*). As *Kakuta, Sasaki & Takishima (1991)* only monitored their samples for 30 min after initial $H_2O_2$ exposure it's possible that full recovery of cilia motility may have occurred in non-surfactant samples at later timepoints. This data along with the current study suggests that respiratory cilia motility can recover if the $H_2O_2$ exposure concentration is low, but higher $H_2O_2$ concentrations comparable with CAM recommendations for $H_2O_2$ nebulizer treatment may cause irreversible impairment to respiratory cilia motility.

### Recovery of cilia generated flow following $H_2O_2$ exposure

The current study also assessed the recovery of cilia generated flow up to 2 h following $H_2O_2$ treatment, which provided one of the more interesting findings of the study. Namely, the study found that while there was recovery of linear flow across the surface of ciliated respiratory epithelia following treatment with the lowest $H_2O_2$ dose (0.1%), as assessed by recovery of microsphere flow directionality; cilia generated flow velocity remained significantly impaired following all $H_2O_2$ treatment doses, even the lowest. This observation highlights an important problem that lies within the published respiratory cilia literature. As mentioned previously, the vast majority of respiratory cilia studies only assess CBF (*Burman & Martin, 1986*; *Feldman et al., 1994*; *Kakuta, Sasaki & Takishima, 1991*; *Kobayashi et al., 1992*; *Nakajima et al., 1999*), probably due to the relatively simple nature of its measurement, and CBF may be fairly meaningless by itself as an assessment for mucociliary clearance.

This study is the first time that recovery of cilia generated flow following $H_2O_2$ treatment was assessed, and clearly shows a disconnect between CBF and cilia generated flow velocity, as highlighted in the 0.1% $H_2O_2$ treatment group. One explanation for this finding is that cilia coordination and/or cilia beat pattern may be impaired by $H_2O_2$ treatment, which significantly reduces their ability to generate flow even if CBF is maintained at normal levels. This hypothesis is supported by previous studies which have shown situations where CBF is maintained (or even elevated), but cilia generated flow is impaired due to defects in cilia beat pattern (*Chilvers, Rutman & O'Callaghan, 2003*; *Raidt et al., 2014*). However, more studies are required to determine if perturbations of cilia coordination and/or cilia beat pattern are caused by $H_2O_2$ treatment.

### Respiratory epithelia $H_2O_2$ cytotoxicity

The cytotoxicity of $H_2O_2$ on respiratory epithelia was assessed 2 h after $H_2O_2$ exposure using fluorescent live/dead staining. Non-$H_2O_2$ exposed control tissues showed a well-organized cobblestone epithelia morphology with a ~3:2 non-ciliated to ciliated cell distribution ratio. $H_2O_2$ treatment caused epithelia morphology to become noticeably disorganized, severity of this disorganization increased with $H_2O_2$ concentration, but no change in non-ciliated to ciliated epithelia cell distribution ratio was observed at any $H_2O_2$ dose suggesting no overall loss of ciliated cells or cilia. A clear dose response effect was seen with cell survival, with increased $H_2O_2$ concentration causing significant increases in epithelia cell death. Most importantly, ciliated epithelial cells appeared considerably more sensitive to the cytotoxic effects of $H_2O_2$ than non-ciliated epithelial cells, with 1% $H_2O_2$ treatment causing death in ~35% of ciliated epithelia cells but only in ~9% of their non-ciliated counterparts.

Three previous studies have reported that $H_2O_2$ is cytotoxic to tracheal epithelia (*Burman & Martin, 1986*; *Kobayashi et al., 1992*; *Nakajima et al., 1999*). These studies included a [51]Cr Cytotoxicity Assay following a 4-h treatment of rat trachea rings with of 3 mM $H_2O_2$ (*Burman & Martin, 1986*); lactate dehydrogenase measurement following 60–90 min treatment of sheep trachea cultures with $10^{-10}$ to $10^{-4}$ M $H_2O_2$ (*Kobayashi et al., 1992*); and a gel electrophoresis DNA fragmentation assay following 24 h treatment

of bovine trachea cultures with 100–1,000 μm $H_2O_2$ (*Nakajima et al., 1999*). The problem with these past studies is that they are all utilized non-specific cytotoxicity assays, and can't directly determine which cells were dying, and while *Nakajima et al. (1999)* suggests from their TEM images that ciliated cells are more sensitive to the cytotoxic effects of $H_2O_2$, no quantification for this was offered (*Nakajima et al., 1999*). Thus, this study clearly demonstrates for the first time that ciliated respiratory epithelial cells are significantly more sensitive to $H_2O_2$ cytotoxicity than their non-ciliated epithelial counterparts.

## CONCLUSIONS

In conclusion, this study demonstrates that a single 10-min $H_2O_2$ exposure, at a concentration based on those recommended by CAM practitioners, results in the significant impairment of respiratory cilia function, as characterized by the complete cessation of all cilia motion and cilia generated flow which does not return after 2 h. Lower $H_2O_2$ concentrations displayed dose response effects, but even the lowest $H_2O_2$ dose studied (0.1%) resulted in significant impairment of cilia beat frequency and cilia generated flow which only partially recovered 2 h following treatment. The toxic effect of $H_2O_2$ was further highlighted by live/dead staining which revealed that ciliated respiratory epithelia cells were significantly more sensitive to $H_2O_2$ induced cell death than their non-ciliated counterparts, as assessed 2 h after a single $H_2O_2$ dose. While this data needs confirmation using *in vivo* models, it suggests that extreme care should be taken when considering treating respiratory epithelia with $H_2O_2$.

## ACKNOWLEDGEMENTS

Author wishes to thank Christine Hall for her sterling technical support, and Serrin Rowarth, Leanne Taylor, Olivia Johnson, Cassandra Bell, and everyone else at the Australian Institute of Tropical Health & Medicine small animal colony for helping source the animal tissues used in this study.

### Funding

The authors received no funding for this work.

### Competing Interests

The authors declare that they have no competing interests.

### Author Contributions

- Richard Francis conceived and designed the experiments, performed the experiments, analyzed the data, prepared figures and/or tables, authored or reviewed drafts of the article, and approved the final draft.

### Ethics

The following information was supplied relating to ethical approvals (*i.e.*, approving body and any reference numbers):

James Cook University Animal Ethics Committee

## Data Availability

The raw data (number of cells with motile cilia, cilia beat frequency, bead tracing for cilia generated flow, and cell counts (dead, ciliated, non-ciliated) measured by immunohistochemistry) is available in the Supplemental File.

## Supplemental Information

Supplemental information for this article can be found online at http://dx.doi.org/10.7717/peerj.14899#supplemental-information.

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
