# Peer review of "The effects of acute hydrogen peroxide exposure on respiratory cilia motility and viability"

_PeerJ, doi:10.7717/peerj.14899_

## Round 0.1 · original submission · Minor Revisions

Reviewers have found some concerns in your manuscript that need to be addressed before considering publishing it.

Please read carefully and attend as soon as possible.

Reviewer 1 ·

Basic reporting

The author of the manuscript performed a series of experiments on ex vivo mouse ciliated epithelia to prove the acute toxicity of hydrogen peroxide on respiratory epithelium cells. The aim of the study was to determine how acute hydrogen peroxide exposure effects the ciliated respiratory epithelia, then to assess the ability of the epithelia to recover from this initial hydrogen peroxide exposure. The author used imaging methods to quantify cilia motility, cilia generated flow, and epithelial damage. It was discovered that cilia motility was compromised after hydrogen peroxide treatment, and the cilia motility can be partly recovered after hydrogen peroxide treatment. But cell death was induced by hydrogen peroxide treatment.
The language of the manuscript is professional, clear, and unambiguous. Sufficient background was provided in the manuscript. The article was well-constructed with sufficient data. The data included supported the conclusion well.

Experimental design

The research question and aim were clearly stated in the article. The experimental detail and information were included in the manuscript. The data supported the conclusion well. The author chose 0.1% to 1% hydrogen peroxide as treatment. The author mentioned this is because this range was recommended within the CAM literature, which is higher than examined in any previous study. However, most of the literature used nebulized hydrogen peroxide which is not identical to the experimental treatment. If the author can offer data about a wider concentration range, it would be more informative to the readers.

In figure 1C and D, it would be better if the author can mention what lines with different colors mean.

Validity of the findings

The article offered proof of the acute toxicity of hydrogen peroxide on the respiratory epithelium. The data are provided and support the conclusion well. The conclusions are well stated and directly linked to the original research question and limited to supporting results.

Additional comments

Although the article lacks deeper mechanistic explanations for the phenomena, this article gives a good insight into the influence of hydrogen peroxide on the respiratory epithelium.

·

Basic reporting

Mucus and other factors could modify this movement and death results in epithelial cells targeting hair cells. It demonstrates the low concentration of H2O2 necessary for the modification of the function of the ciliated cells of the epithelium of the airway, it even demonstrates the modification of the epithelium with death of. Taking into account recommendations made by complementary and alternative medicine (CAM) and demonstrating with previously standardized methods, the toxicity of H2O2 "inhalation therapy" is demonstrated.

Age of the mice could influence the results obtained: Bailey KL, et al., Aging causes a decrease in the frequency of ciliary beats, mediated by PKCε. I am J Physiol Lung Cell Mol Physiol. 2014 March 15; 306(6):L584-9. doi: 10.1152/ajplung.00175.2013 (study using mice)

line 288 the figure referring to flow velocity is 2C and flow directionality is 2D modify the wording to avoid being redundant in the information, for example:
--Linear flow also recovered in the 0.2% and 0.5% H2O2 treatment groups but remained significantly lower compared to control values (P<0.0001) following 0.2% H2O2 treatment, and significantly lower compared to both control (P<0.0001) and 0.1% H2O2 treatment (P<0.01) following 0.5% H2O2 treatment (Figure 2D) -- or mention repeatedly the figure that it presents in the same paragraph

line352 with respect to H2O2 is demonstrated with the Brownian movement of the sphere, however the movement of the cilium would not necessarily reflect a specific linear movement, as mentioned in the previous article.

Experimental design

4 trachea samples from each mouse and 1 shamp (what substance was used as shamp?) and 4 with different concentrations of H2O2 treatment were randomized for each group.

In the determination of directionality, they assume linear movement, and Brownian for waiting in movement, in the case of the group without intervention, movement, speed and directionality are demonstrated, however, said movement was not taken into account at the beginning of the intervention. with H2O2. Line 161-167

Line 242 sham and 1% H2O2 have the same viscosity or fluid density, or this can modify the motility and speed?

Validity of the findings

Line 232 is the SEM shown correct data? Mean of 34.1 is equal to SEM 34.1%? and in the 0.1% H2O2 group the mean is 5.4 with a SEM of 7.3%.

Line 281 why a linear or straight movement is assumed in the microsphere?, there is another model of movement that can be raised, including a stagnation or deposition of the molecule(s), which could be influenced by the speed of the flow or the zero speed of the flow.


Line 286 same comment, at zero time there is no movement due to the absence of the tissue, the control behaves completely differently.
from 0.1% to 1% behaves in the same way

line 336 there are different conditions that could modify the movement, and the ciliary flow, for this reason my question about the density and viscosity of the media in which the movement of the microspheres was evaluated

Additional comments

Line 411, so if the concentration is changed to a very low of H2O2 0.1% by CAM, would its use be recommended?

---

## Round 0.2 · accepted · Accept

Thank you very much for attending to the previous concerns of the reviewers.
I have read it, and your manuscript can be accepted with the current modifications.

In my opinion, your manuscript is ready for publication.